# Comparative Analysis of Stock Bubble in S&P 500 Individual Stocks: A Study Using SADF and GSADF Models

**Durga Acharya**

College of Business, Westcliff University, Irvine, CA 92614, USA; d.acharya135@westcliff.edu

**Abstract:** Stock bubbles are characterized by unpredictable price surges and subsequent declines, causing significant losses for investors. This study investigates the effectiveness of the Generalized Sup Augmented Dickey–Fuller (GSADF) test in identifying mild explosive patterns and speculative bubbles within individual S&P 500 stocks, as compared to the Sup Augmented Dickey–Fuller (SADF) test. Utilizing real-time monitoring data, this research examines unit roots, stationarity, and the ability to detect multiple structural breaks. The GSADF test consistently outperforms the SADF test in rejecting the null hypothesis, demonstrating greater sensitivity and efficacy in recognizing stock bubbles. Monte Carlo simulations address size distortions in the GSADF test, enhancing accuracy.

**Keywords:** Augmented Dickey–Fuller test (ADF); bubble; stationarity; unit roots; volatility





## 1. Introduction

A stock bubble, also known as an asset or speculative bubble, garners substantial attention from academia and practitioners. According to Karimov (2017), a stock bubble occurs when asset prices transiently accelerate upward over and above their fundamental or intrinsic value. This increase in prices is driven by new speculators seeking to profit from even higher prices, rather than the fundamental value of the asset itself (French 1991). Various factors, such as increased investor enthusiasm and speculation, media hype, analyst recommendations, and other forms of positive sentiment, can contribute to the rapid surge in stock prices, leading to a self-fulfilling cycle of buying and selling. Furthermore, the formation of a stock bubble can also be influenced by low interest rates and favorable credit conditions. These conditions make it easier for investors to access credit, which can be invested in stocks, thereby augmenting the demand and driving up the price.

Asset price bubbles are often blamed for causing economic recessions. According to Aliber and Kindleberger (2015), such bubbles are linked to a state of economic optimism that can contribute to a subsequent decline in economic activity. They further suggest that the failure of financial institutions during these bubbles can disrupt the channels of credit, causing a slowdown in economic recovery (p. 134).

Stock market crashes and economic recessions share a positive relationship, creating a self-perpetuating cycle. When a stock bubble bursts, entrepreneurs' and investors' net worth declines, leading to reduced credit and investment. This, in turn, lowers labor demand, causing wage declines. Due to nominal wage rigidity, institutions may implement minimum working hours or rationing, resulting in involuntary unemployment. The rise in unemployment can disrupt the intertemporal allocation of resources, ultimately leading to a recessionary phase (Biswas et al. 2020). The relationship between unemployment and return on capital investment is contrastive, leading to a decline in the net worth of investors. Entrepreneurs' ability to invest depends on their net worth, and a net worth decline can reduce future capital stock. This, in turn, exacerbates the downward pressure on labor demand and perpetuates a self-reinforcing cycle, which continues until the capital stock falls sufficiently to reach a bubble-less steady-state equilibrium (Biswas et al. 2020). The complex interplay of these factors underscores the potential consequences of asset price bubbles and their harmful effects on economic stability.

Retail investors who engage in stock trading during a speculative bubble face the peril of substantial financial losses when the bubble eventually bursts, resulting in a precipitous decline in stock prices (Emmons and Noeth 2012). This risk is particularly pronounced for individual investors who may lack the requisite expertise to conduct thorough evaluations of stock fundamentals and accurately assess the risks inherent in bubble-related investments. Furthermore, investing in a stock bubble can result in opportunity costs for individual investors, as they may become overly focused on the stock market, neglecting other asset classes, such as bonds or real estate. This oversight can hinder their ability to diversify their investment portfolio and explore alternative investment avenue options.

Due to the nature of it, stock prices often exhibit rapid and unpredictable fluctuations. When prices surge significantly beyond their actual value, resembling a bubble, this can typically be recognized and studied retrospectively after a price decline occurs. In such instances, both institutional and individual investors may experience substantial wealth losses over the different asset classes they hold (Brunnermeier and Oehmke 2012).

Furthermore, when investors hold unrealistic expectations of perpetual demand and profitability in a specific stock, it fosters irrational exuberance, driving the price far above its intrinsic value and the company's actual potential. Positive sentiment about a stock's future profitability can lead to its current price exceeding its fundamental value. When this disparity arises, it suggests the presence of a stock bubble (Shiller 2000; Stiglitz 1990). Unfortunately, retail investors, often unaware of these dynamics, may incur substantial losses when the bubble inevitably bursts.

Detecting and managing stock bubbles is challenging due to their unpredictability (Focardi and Fabozzi 2014). Investors struggle to accurately estimate peak prices and bubble duration, often remaining invested and continuing to invest despite overvaluation. Unfortunately, once a stock enters a bubble zone, investors are unable to divest themselves of it before it collapses, leaving them exposed to significant financial losses. The lack of understanding of stock bubble formation and dynamics poses risks to both retail and hedge fund investors, potentially favoring short-term gains over long-term strategies.

The bursting of a stock bubble erodes investor confidence and trust in the stock market as a secure investment. Investors purchasing stocks at inflated prices often face significant declines, causing frustration and mistrust. For instance, retirees, reliant on stocks for retirement income, may suffer financial insecurity if a significant portion of their portfolio depreciates during a bubble, lacking the capacity for recovery through additional investments or employment.

Researchers have made significant attempts to develop the econometric technique to detect the indication of the existence of bubbles. The study of equity market bubbles, particularly in the United States, has garnered significant interest. However, existing methodologies for detecting bubbles are still insufficient in providing definitive evidence for the bubble hypothesis. During the earlier phase, Shiller (1981) examined stock price volatility and its relationship to dividend fluctuations. He found that stock prices exhibited excessive volatility due to the introduction of new information about future dividends. Shiller argued that this high volatility could not be solely attributed to dividend changes, even when accounting for uncertainty in dividend growth. His research provided evidence that stock prices moved excessively, suggesting market inefficiencies and potential bubbles (Shiller 1981).

West (1984) utilized Shiller's data to detect stock market bubbles using a three-step test based on the Euler equation and AR representation. In a separate study, West (1987) employed a specification test to confirm the presence of a stock market bubble by linking dividend patterns to equilibrium stock prices. However, Camerer (1989) questioned West's use of a constant discount rate and suggested further investigation into discount rate predictability.

Diba and Grossman (1988) examined the presence of an explosive rational bubble in stock prices, using a model that considered market fundamentals as a combination of unobservable variables, including the present value of anticipated dividends at a constant

discount rate. Their study found that stock prices and dividends were non-stationary before differencing but became stationary after the first differencing. However, they did not find evidence of cointegration between them, leading to the conclusion that there was no support for the existence of an explosive rational bubble in stock prices. They argued that if a bubble were to exist in the current market, it would have always existed, and once a bubble reaches zero, it cannot reappear.

Furthermore, Evans (1991) challenged the theory proposed by Diba and Grossman by introducing a novel model for periodically collapsible bubbles. Evans's model suggests that real stock price bubbles cannot have negative values and that, contrary to Diba and Grossman's argument, if a bubble were to reach zero, it could re-emerge. Similarly, Froot and Obstfeld (1991) contributed significantly to the bubble theory with an intrinsic bubble model that refines the concept of the rational bubble. Their model distinguishes between exogenous fundamental determinants of asset prices and extraneous variables, which can influence the self-fulfilling expectations process of the bubble.

As far as time series-based models are concerned, time series-based analysis is a common approach for studying speculative bubbles in financial markets, but empirical findings have lacked consistency. Numerous formal methods exist to assess data stationarity, with the Dickey–Fuller test (Dickey and Fuller 1979) being a prominent one. This test is noteworthy as it allows for the examination of non-stationarity, akin to detecting unit roots. Many studies have utilized the Dickey–Fuller test to identify speculative bubbles, and this summary provides an overview of these investigations. Craine (1993) introduced a time-series model and employed the standard augmented Dickey–Fuller (ADF) test to evaluate the stationarity of the log dividend–price ratio using S&P 500 data from 1876 to 1988.

Cuñado et al. (2005) analyzed the NASDAQ stock market index from 1994 to 2003 for a rational bubble using fractional methodology. They found mixed results: no bubble with monthly data but evidence for a bubble with daily and weekly data. Similarly, Koustas and Serletis (2005) used the ARFIMA method to study the S&P 500 log dividend yield. Their results supported the presence of a rational bubble in stock prices, rejecting the null hypothesis of no bubble.

Phillips and Yu (2011) proposed a novel approach to detecting explosive bubbles using sequential unit root tests. Their method, called the sup augmented Dickey–Fuller (SADF) test, is designed to identify the start and end dates of a single explosive bubble and is considered superior to existing bubble detection tests. This approach builds upon previous work by Diba and Grossman (1988) who suggested that no bubbles exist in the S&P 500. Phillips and Yu (2011) introduced a new recursive regression methodology that overcomes this limitation and provides consistent dating of bubble origination and collapse.

However, the SADF test may have reduced statistical power and yield inconsistent results when dealing with multiple bubble formations and collapses in the sample period. To address this limitation, Phillips et al. (2015a) introduced an extension called the generalized sup ADF (GSADF) method. This approach uses a recursive backward regression technique to precisely identify bubble origination and termination dates while accounting for multiple exuberance and collapse episodes. The GSADF method represents an improvement over the earlier approach, providing more robust and reliable bubble identification within the sample period.

The purpose of this study is to identify the presence of mildly explosive patterns and bubbles in individual stocks that are listed in the S&P 500 stock. This study aims to use real-time monitoring information on selected stocks to evaluate their behavior when they have a bubble component. This will involve identifying the beginning and end periods of the speculative bubble in the stock. To achieve this, this study intends to apply the sup augmented Dickey–Fuller unit root test (SADF) and generalized sup augmented Dickey–Fuller unit root test (GSADF), which were introduced by Phillips and Yu (2011) and Phillips et al. (2015a), respectively, for individual stocks.

## 2. Data and Methodology

This section provides a comprehensive overview of the sampling methodology, data collection, and analytical technologies adopted to execute this study.

### 2.1. Data and Sample

This quantitative study examines stock growth within the S&P 500 index by applying specific selection criteria. Stocks meeting the criteria include those with at least a 10% price increase on any trading day during the five-year and one-quarter period from 1 January 2018 to 31 March 2023. Such stocks are deemed to potentially exhibit explosive behavior.

Additionally, this study addresses the impact of trading volumes on stock prices. Low-volume stocks, characterized by limited liquidity due to fewer buyers and sellers, are subject to greater price volatility. To mitigate this effect, this study selects stocks with a minimum of one million trades per day, emphasizing higher liquidity and price stability. This approach aims to avoid distortions caused by low-volume stocks and focuses on identifying persistent price bubbles. The chosen five-year and one-quarter time frame provides insight into multiple market cycles, aiding in the detection of bubble patterns and trends. Daily data analysis is crucial as it allows for the identification of subtle trends and patterns that may be missed in weekly or monthly data.

A Python script was used to screen S&P 500 stocks, downloading data and identifying those meeting criteria (10% price change, ≥1 million daily trades). This approach identifies potential bubble candidates. Similarly, R software (R 4.3.2 binary for macOS 11), known for its user-friendliness, employs the "exuber" package to analyze structural breaks and explosive behavior, as demonstrated in prior studies (Pavlidis et al. 2019), and is being used to obtain the explosiveness of the selected samples.

### 2.2. Methodology

This study employed two methods to detect the explosive pattern in the individual stock: the supremum augmented Dickey–Fuller (SADF) test proposed by Phillips and Yu (2011) and the generalized supremum augmented Dickey–Fuller (GSADF) test proposed by Phillips et al. (2015a).

#### 2.2.1. SADF Approach

The study by Phillips and Yu (2011) introduced a novel method that is capable of detecting the periodic collapsing bubbles identified in the work of Evans (1991). The researchers conducted extensive simulation studies and developed a right-tail Dickey–Fuller test, which can identify the originating and collapsing dates of a bubble with greater power than the cointegration methodology. The testing procedure involves the use of a sup augmented Dickey–Fuller (SADF) method to identify the presence of explosive behavior in stock prices. Specifically, for each time series ($x_1$), the ADF test for a unit root against the alternative of an explosive root (right-tailed) is employed. The autoregressive specification of $y_t$ is estimated by the least square method as shown below.

$$y_t = \mu + \delta y_{t-1} + \sum_{i=1}^{k} \phi_j \triangle y_{t-i} + \varepsilon_t, \tag{1}$$

where $y_t$ is a logarithm; $k$ is the transient lag order; $\triangle y_{t-i}$ with $i = 1, \ldots, k$ are lagged first differences of the series included to accommodate the serial correlation, $\varepsilon_t \sim IID\left(0, \delta^2_{r1,r2}\right)$; and $\mu$, $\delta$, and $\phi_j$ are the regression coefficient with $i = 1, \ldots, k$. The objective is to test the unit-root null hypothesis $H_0 : \delta = 1$ versus the right-tailed alternative of explosiveness, $H_1 : \delta > 1$. The above equation is employed repeatedly using a subset of the sample data with one additional observation at each pass in the forward recursive regression. The SADF test uses a rolling window approach to test for the presence of a unit root in a time series. At each step, a subset of the data within the rolling window is used to estimate the autoregressive process. The SADF test then computes a test statistic based on the residuals

of the estimated model and compares it to a critical value to determine if the null hypothesis of a unit root can be rejected.

The number of lags in the autoregressive process can have a significant impact on the performance of the SADF test. Too few lags can lead to underfitting, while too many lags can lead to overfitting. To determine the optimal number of lags, forward recursive regression is used to select the lag length that maximizes the test statistic.

Let $r_w$ be the window size of the regression. The window size $r_w(r_w = r_2 - r_1)$ extends from $r_0$ to $r_1$, where $r_0$ is the smallest sample window width fraction and 1 is the largest window fraction. The starting point $r_1$ is fixed at 0, and the ending point for each sample $r_2$ is equal to $r_w$ and changes from $r_0$ to 1 (Phillips et al. 2015a, p. 1048). The ADF statistics for a sample that runs from 0 to $r_2$ is denoted by $ADF_0^{r_2}$. The SADF statistic is defined as the sup value of the ADF statistic sequence, which is

$$SADF(r_0) = r_2 \in [r_0, 1] sup ADF_0^{r_2} \tag{2}$$

The Phillips and Yu (2011) SADF test statistic is defined as a sup value of the sequence of $ADF_0^{r_2}$. Under the I(1) null, the limit distribution of the $SADF(r_0)$ statistic is given by

$$r_2 \in [r_0, 1] sup. \frac{\int_0^{r_2} \widetilde{w}(r) d\widetilde{w}(r)}{\left( \int_0^{r_2} \widetilde{w}(r)^2 dr \right)^{\frac{1}{2}}} \tag{3}$$

where $\widetilde{w}$ is a demeaned Wiener process (Brownian motion). Whenever $SADF(r_0)$ exceeds the corresponding right-tailed critical value from its limit distribution, the unit root hypothesis is rejected in favor of mildly explosive behavior.

$$\hat{r}_e = r_2 \in [r_0, 1] inf \left\{ r_2 : ADF_{r_2} > cv_{r_2}^{adf} \right\} \tag{4}$$

$$\hat{r}_f = r_2 \in [\hat{r}_e, 1] inf \left\{ r_2 : ADF_{r_2} < cv_{r_2}^{adf} \right\} \tag{5}$$

The SADF test statistic cannot locate the beginning and end date of the bubble. In order to identify the beginning and collapse date, a recursive test statistic $ADF_r$ versus the right-tailed critical value $(cv_{r_2}^{adf})$ needs to be compared. If $r_e$ is the beginning date and $r_f$ is the collapse date, the estimate of these dates can be constructed as below.

2.2.2. Generalized Supremum Augmented Dickey–Fuller (GSADF) Test

Phillips et al. (2015a) proposed a novel approach to enhance the detection capability of multiple stock-price bubbles through a recursive (right-tailed) unit root test called the generalized SADF (GSADF) test. The GSADF test is built on the same principles as the SADF test but employs more subsamples for estimation than the $SADF(r_0)$ test by allowing greater flexibility in the selection of starting points for subsamples $(r_1)$. This added degree of flexibility in the estimation window of the GSADF test leads to increased statistical power, enabling it to identify multiple and periodically collapsing episodes of explosiveness, whereas the SADF test is limited to identifying only a single episode.

Phillips et al. (2015a) derive the asymptotic null distribution of the SADF and GSADF tests' statistics on the basis of the prototypical model with a weak (local to zero) intercept form.

$$H_0 : y_t = dT^{-\eta} + y_{t-1} + \varepsilon_t, \ \varepsilon_t \sim i.i.d(0, \sigma^2) \tag{6}$$

where $d$ is a constant, $T$ is the sample size, and the parameter $\eta$ is a localizing coefficient that controls the magnitude of the intercept and drift as $T \to \infty$. By solving the equation, $y_t = d\frac{t}{T^{\eta}} + \sum_{j=1}^{t} \varepsilon_j + y_0$, showing the deterministic drift $d\frac{t}{T^{\eta}}$. When $\eta > 0$, the drift is small relative to a linear trend; when $\eta > 1/2$, the drift is small relative to the martingale component of $y_t$; and when $\eta < 1/2$, the standardized output $T^{-1/2}y_t$ acts asymptotically like a Brownian motion with drift.

The alternative hypothesis for the mildly explosive process is

$$H_1 : y_t = \delta_T Y_{t-1} + \varepsilon_t, \tag{7}$$

where $\delta_T = 1 + cT^{-\theta}$ with $c > 0$ and $\theta \in (0,1)$. Under null I(1), the limit distribution of the GSADF statistic is given by

$$r_1 \in [0,\, r_2 - r_0], r_2 \in [r_0, 1] \sup \left\{ \frac{\frac{1}{2} r_w \left[ w(r_2)^2 - w(r_1)^2 - r_w \right] - \int_{r_1}^{r_2} w(r) dr [w(r_2) - w(r_1)]}{r_w^{\frac{1}{2}} \left\{ r_w \int_{r_1}^{r_2} w(r)^2 dr - \left[ \int_{r_1}^{r_2} w(r) dr \right]^2 \right\}^{\frac{1}{2}}} \right\} \tag{8}$$

where $r_w = r_2 - r_1$ is the size of the expanding window. Whenever $GSADF(r_0)$ exceeds the corresponding right-tailed critical value from its limit distribution, the unit root hypothesis is rejected in favor of mildly explosive behavior.

The GSADF statistic's limit distribution in the above equation is equivalent to the scenario where the regression model incorporates an intercept, and the null hypothesis is a random walk or a unit root process without drift. The standard limit distribution of the ADF statistic is a special case of (8) with $r_1 = 0$ and $r_2 = r_w = 1$. In contrast, the limit distribution of the single recursive SADF statistic is a further specific with $r_1 = 0$, and $r_2 = r_w = \in [r_0, 1]$.

The distribution of the asymptotic GSADF test depends on the choice of the smallest window size, denoted as $r_0$. In practical applications, $r_0$ must be carefully selected based on the total number of observations, $T$. If $T$ is relatively small, it is important to set $r_0$ to a large enough value to ensure adequate initial estimation. Conversely, if $T$ is large, $r_0$ can be set to a smaller value to maximize the ability to detect an early explosive episode.

However, it is important to note that the theoretical framework for break-test methodology requires $r_0$ to be bounded away from zero as $T$ approaches infinity. To address this, extensive simulations and recommendations for a rule for selecting $r_0$ based on a lower bound of 1% of the full sample and a convenient functional form of $r_0 = 0.01 + 1.8/\sqrt{T}$ are applied (Phillips et al. 2015b).

## 3. Characteristics of the Data

This research utilized data from Yahoo Finance, focusing on Standard & Poor's 500 (S&P 500) companies from January 2018 to March 2023. The S&P 500 is a critical stock market index monitoring 500 major publicly traded U.S. companies and covers about 80% of the total U.S. equity market. Analyzing stocks across diverse sectors within the S&P 500 index offers a robust framework for garnering insightful perspectives on the dynamics of stock price fluctuations in varied market conditions. The research has opted for US-based stocks over other international counterparts due to the facile accessibility of data for comprehensive data collection.

The time frame is considered sufficient to capture a significant number of bubbles in individual stocks. This particular timeframe was chosen as it is well-suited to the goal of the study, which is to test individual stocks for bubbles. It is noteworthy that individual stocks are characterized by relatively shorter bubble episodes compared to the overall market, and thus, the inclusion of long-term price history is deemed unnecessary. Moreover, the chosen timeframe encompasses a diverse range of market conditions, including periods of economic growth and recession. For instance, the ongoing COVID-19 pandemic led to a recession in 2019, which is also covered within this timeframe.

In this study, a probability sampling method was employed, which involved the selection of individual stocks that demonstrated price increases exceeding 10% on any given trading day during the five-year and one-quarter period, and maintained a minimum daily trading volume of one million shares. A Python code is used to automate this selection process, resulting in a sample size of 104 companies for further analysis and distribution across sectors, presented in Figure 1.

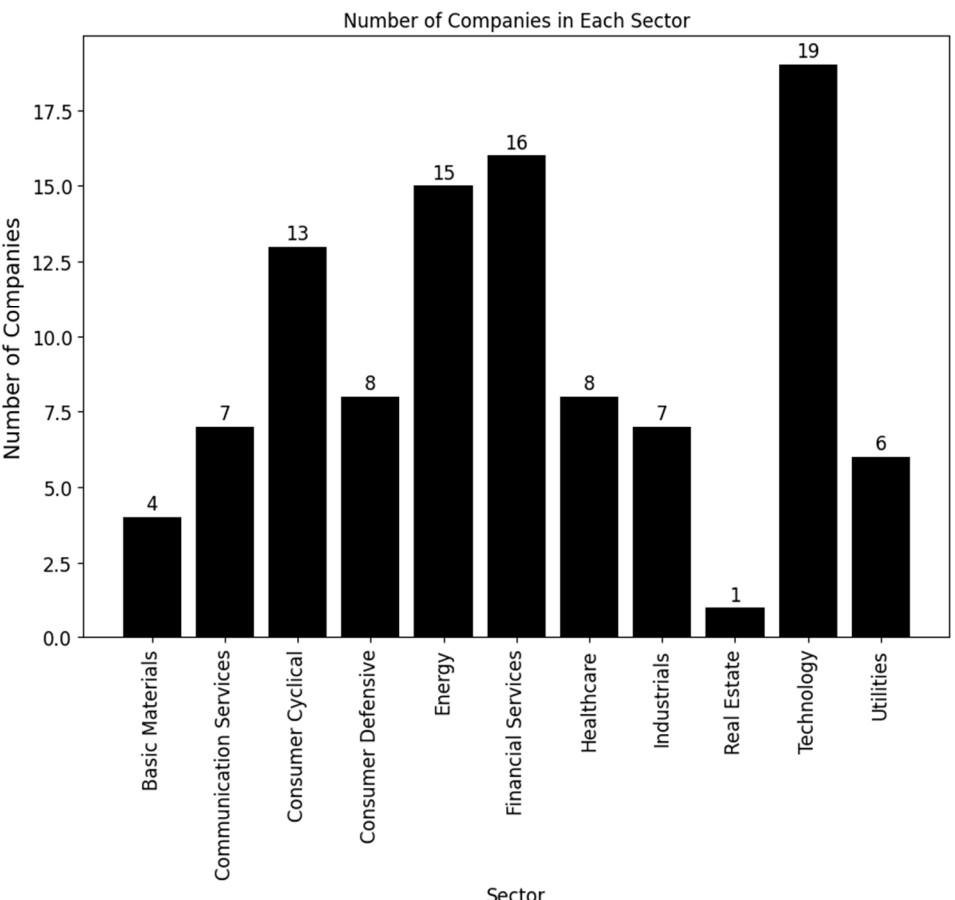

**Figure 1.** Sectors with the number of companies.

Figure 1 illustrates the distribution of companies across various industry sectors. The Technology sector boasts the largest number of companies (19), followed by the Financial Services sector (16) and the Energy sector (15). Conversely, the Real Estate, Basic Materials, and Utilities sectors each comprise only one, four, and six companies, respectively.

Table 1 categorizes companies into three trading volume groups: Low Volume (13 companies, lowest 25%), Medium Volume (67 companies, middle 50%), and High Volume (24 companies, top 25%).

**Table 1.** Number of companies by trading volume groups.

| Group | Number of Companies |
|---|---|
| Low Volume | 13 |
| Medium Volume | 67 |
| High Volume | 24 |

Market capitalization, another crucial metric, is the total value of a company's outstanding shares, calculated by multiplying outstanding shares by the market price as of 31 March 2023.

Table 2 displays the distribution of companies by market capitalization as of 31 March 2023, organized into distinct segments. The largest segment comprises 20 mega-cap companies with market values exceeding USD 200 billion. Following this, 82 large-cap companies are noted with market values ranging from USD 10 billion to USD 200 billion. There are only two companies in the mid-cap segment, possessing market values between USD 2 billion and USD 10 billion, while the small-cap segments contain no listed companies.

**Table 2.** Number of companies by market capital.

| Group | Number of Companies |
|---|---|
| Mega-cap companies | 20 |
| Large-cap companies | 82 |
| Mid-cap companies | 2 |
| Small-cap companies | 0 |

Since this research examined 104 stocks, and due to the limited space availability, it showcases the top and bottom ten companies, selected based on their highest percentage change in stock prices during the sample periods. Importantly, even the companies with the lowest percentage increases in this segment still adhered to this study's requirement of a minimum 10% increase in stock prices.

Table 3 presents data on ten companies' stock prices, highlighting their minimum and maximum price points, highest percentage changes, and corresponding dates. These companies are the top ten performers in terms of percentage increase during the sample period, with stock prices ranging from USD 3.80 to USD 9.04 (minimum) and USD 22.25 to USD 76.45 (maximum), and percentage changes ranging from 29.46% to 74.59%.

**Table 3.** Top ten companies by highest growth.

| Stock | Minimum Price | Maximum Price | Highest Daily Jump (%) | Date of Highest Change |
|---|---|---|---|---|
| PCG | 3.8 | 48.96 | 74.593 | 23 January 2019 |
| AAL | 9.04 | 56.989 | 41.097 | 3 June 2020 |
| CCL | 6.38 | 66.218 | 39.291 | 6 November 2020 |
| BBWI | 7.184 | 76.459 | 39.04 | 23 March 2020 |
| EQT | 4.835 | 49.957 | 37.32 | 12 March 2020 |
| OXY | 8.764 | 75.595 | 33.698 | 4 June 2020 |
| MGM | 7.133 | 50.35 | 33.115 | 23 March 2020 |
| HST | 8.7 | 20.426 | 30.029 | 6 November 2020 |
| FCX | 5.149 | 50.851 | 29.685 | 23 March 2020 |
| APA | 3.854 | 50.293 | 29.466 | 23 March 2020 |

Note. The data were collected by the author in May 2023.

Table 4 showcases the ten lowest-performing stocks within a larger group of growth stocks. These stocks had the highest single-day percentage increases in stock prices, but their gains were comparatively smaller than those of other stocks in the growth sample. For instance, Altria Group, Inc. (MO), Richmond, VA, USA had a 10.02% increase on 13 March 2020, marking its highest single-day percentage gain during the chosen period, but it was the lowest among the listed stocks. Notably, eight of these ten companies experienced their highest daily percentage returns in 2020, while one achieved its highest return in 2019, and another in 2021. Figure 2 below shows the number of stocks with their highest price increase each year between 2018 and 2023.

Figure 2 illustrates the evolution in the number of stocks experiencing their highest single-day percentage growth from 2018 to 2023. There is a notable increase in such occurrences over the selected period. In 2018, only three stocks exhibited this level of growth, which expanded to seven in 2019 and surged to eighty-six in 2020. However, the data for 2021 and 2022 show a decline in the number of stocks with this level of growth. In 2021, only three stocks matched the 2018 count, and in 2022, there were just four. By 2023, this trend further decreased to only one stock.

**Table 4.** Lowest ten companies by lowest growth.

| Stock | Minimum Price | Maximum Price | Highest Daily Jump (%) | Date of Highest Daily Jump |
|---|---|---|---|---|
| MO | 24.18 | 52.071 | 10.022 | 12 March 2020 |
| T | 14.22 | 22.601 | 10.022 | 12 March 2020 |
| PM | 50.949 | 104.741 | 10.035 | 12 March 2020 |
| MDT | 67.067 | 128.55 | 10.189 | 23 March 2020 |
| GOOG | 48.811 | 150.709 | 10.449 | 25 July 2019 |
| PFE | 23.971 | 58.154 | 10.855 | 4 November 2021 |
| CVS | 46.248 | 107.334 | 10.899 | 16 March 2020 |
| ABT | 51.583 | 137.81 | 10.936 | 23 March 2020 |
| MDLZ | 33.605 | 69.985 | 11.281 | 23 March 2020 |
| IBM | 76.9 | 146.742 | 11.301 | 23 March 2020 |

Note. Author's own calculation.

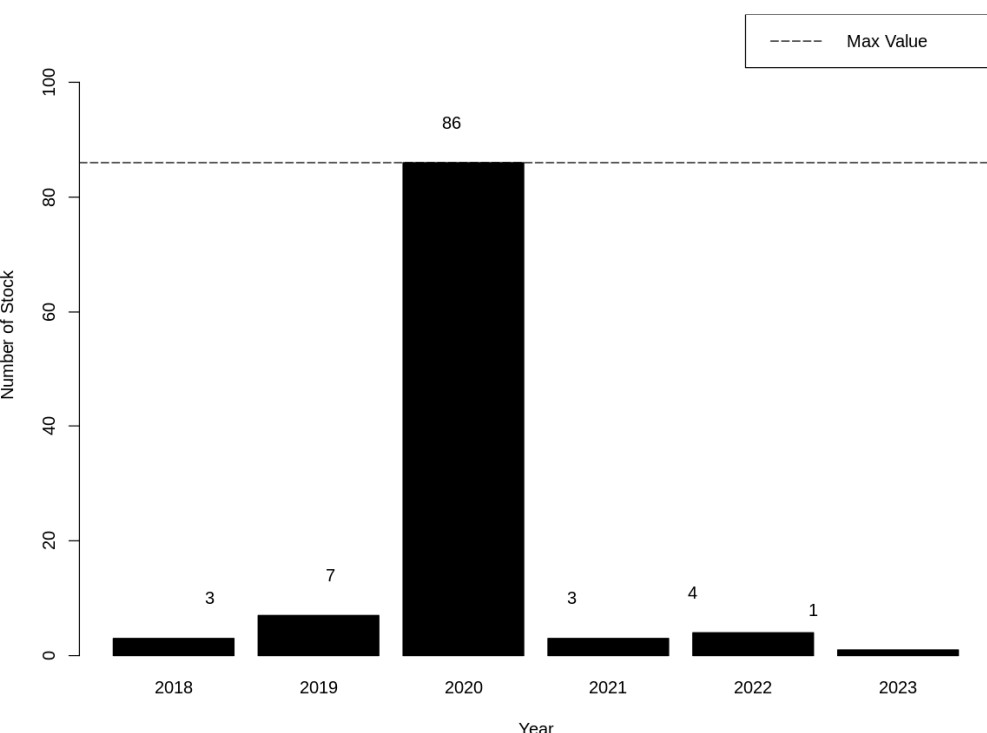

**Figure 2.** Number of stocks with highest price increase by year (2018–2023).

## 4. Results and Discussion

To determine the smallest window size ($r_0$) for the GSADF test, it is crucial to consider the total number of observations (T). For practical purposes, a rule of thumb, based on research by Phillips et al. (2015a), recommends setting $r_0$ to a lower bound of 1% of the full sample using the formula $r_0 = 0.01 + 1.8/\sqrt{T}$.

In our study with 1320 observations spanning from 1 January 2018 to 31 March 2023 (1320 trading days), we followed this recommendation, resulting in an $r_0$ value of 0.06. This corresponds to a minimum moving window of 6% or a minimum window size of 78 observations. The test began with the first $r_0$ observations in the time series.

Regarding bubble duration, Phillips et al. (2015b) found that it should exceed a parameter δ log (T), which grows proportionally with the dataset's size. For our dataset covering 1320 days, the optimal minimum duration for bubble identification is 7 days. Table 5 below shows some of the dominant variables for the test.

**Table 5.** Dependent variables for recursive test.

| Variable | Value |
|---|---|
| Number of Observations | 1320 |
| Minimum Window Size | 78 |
| Minimum Duration | 7 |
| Optimum Lag | 12 |

For the optimal lag size, the Akaike Information Criterion (AIC) method was used. Importantly, the optimal lag size for each of the 104 stocks, each with 1320 observations, was determined to be 12.

### 4.1. Effectiveness of GSADF Test

This study aims to assess the GSADF test's effectiveness in identifying explosive patterns, which are sudden and substantial price movements in individual stocks. By applying the GSADF test to a dataset of individual stock prices, the research evaluates its reliability in detecting these patterns.

The GSADF test calculates a test statistic measuring the maximum deviation of the time series from a unit root process across various possible breakpoints. Specifically, for each time series, the GSADF test analyzes overlapping windows of 78 observations with a lag size of 12. If the test statistic surpasses the critical value, the null hypothesis of a unit root is rejected.

In simpler terms, rejecting the null hypothesis in the GSADF test implies that the time series has undergone significant structural changes or breaks. In this context, it suggests that current stock prices may not be sustainable. Table 6 presents the results of the ADF, SADF, and GSADF tests' statistics for the selected stocks.

**Table 6.** The test statistics of ADF, SADF, and GSADF tests.

| Stock | ADF | SADF | GSADF |
|---|---|---|---|
| AAL | −2.943 | −0.693 | 2.684 * |
| AAPL | −0.708 | 3.051 * | 3.080 * |
| ABBV | 0.035 * | 2.053 * | 2.929 * |
| ABT | −1.682 | 0.855 | 1.944 |
| AIG | −1.941 | 0.354 | 4.449 * |
| AMAT | −0.759 | 3.058 * | 3.074 * |
| AMD | −1.373 | 2.780 * | 2.801 * |
| AMZN | −1.732 | 1.657 * | 2.035 |
| APA | −1.998 | −0.625 | 1.707 |
| ATVI | −1.437 | 0.074 | 2.266 |

Note. Table 6 demonstrates the test statistics of the first ten companies. * Denotes the rejection of the null hypothesis at a 95% significance level.

Table 6 provides test statistics for various methodologies applied to the initial ten companies. Nonetheless, a direct comparison of the test statistics alone is inadequate without considering the corresponding critical values for each of the stocks. To address this, this study employed a Monte Carlo simulation approach to generate these critical values for comparison. The Monte Carlo simulation technique involves generating numerous random time series with statistical properties resembling the original data. These simulated time series are then employed to estimate the distribution of the test statistic under the null hypothesis of a unit root, indicating that the time series data are non-stationary.

To establish critical values for the GSADF test applied to 104 stocks, I initially estimated model parameters, as displayed in Table 6. The Monte Carlo simulation is used to generate numerous synthetic time series mimicking the statistical properties of the chosen time series. For each synthetic series, I computed and documented GSADF test statistics. Subsequently,

I derived critical values for the GSADF test at a 95 percent significance level from the distribution of these simulated test statistics.

The critical values for the ADF, SADF, and GSADF tests at different significance levels (90%, 95%, and 99%) were obtained through the Monte Carlo simulation and are presented in Table 7.

**Table 7.** The critical value of Monte Carlo simulation of ADF, SADF, and GSADF.

| Significance Level | ADF | SADF | GSADF |
|:---:|:---:|:---:|:---:|
| 90 | −0.420 | 1.302 | 2.157 |
| 95 | −0.077 | 1.543 | 2.387 |
| 99 | 0.513 | 2.157 | 2.813 |

Datestamp Result

Identifying the start and end dates of explosive behavior in stock prices depends on the minimum duration of exuberance. In this context, an origin date is established when the time series of recursive test statistics $GSADF_r$, characterized by r values within the range of $[r_0, 1]$, surpasses the critical value associated with those statistics. Similarly, a termination date is designated when the critical value of $GSADF_r$ with $r$ values spanning $[r_0, 1]$ exceeds the corresponding test statistics. Phillips and Shi (2020) noted that the origination of a bubble or crisis episode is determined as the point when the GSADF test statistic first surpasses its critical value, while the termination date corresponds to the point when the supremum test statistic subsequently falls below its critical value, establishing two distinct stopping times for the episode.

The data in Table 8 reveal several noteworthy observations. For example, Apple Inc. (AAPL), Cupertino, CA, USA experienced a peak on 11 November 2019, following an upward trend that began on 21 October 2019. This peak persisted for 29 days, concluding on 2 December 2019. Similarly, AbbVie Inc. (ABBV), North Chicago, IL, USA reached its highest point on 15 November 2019 after a period of upward movement starting on 8 November 2019. This peak lasted for 8 days, ending on 20 November 2019. Additionally, American International Group Inc. (AIG), New York, NY, USA and Applied Materials Inc. (AMAT), Santa Clara, CA, USA encountered peaks with varying durations. Table 8 also indicates the presence of multiple episodes of bubbles within each stock, characterized by different duration dates.

**Table 8.** Bubble result of the stocks based on GSADF test.

| Stock Name | Start Date | Peak Date | End Date | Duration (Days) |
|:---:|:---:|:---:|:---:|:---:|
| AAPL | 21 October 2019 | 11 November 2019 | 2 December 2019 | 29 |
| AAPL | 26 December 2019 | 13 January 2020 | 31 January 2020 | 24 |
| AAPL | 3 August 2020 | 1 September 2020 | 8 September 2020 | 25 |
| ABBV | 8 November 2019 | 15 November 2019 | 20 November 2019 | 8 |
| ABBV | 15 March 2022 | 8 April 2022 | 13 April 2022 | 21 |
| AIG | 9 May 2019 | 20 May 2019 | 31 May 2019 | 15 |
| AMAT | 11 February 2021 | 24 February 2021 | 4 March 2021 | 14 |
| AMAT | 9 March 2021 | 17 March 2021 | 18 March 2021 | 7 |
| AMAT | 22 March 2021 | 5 April 2021 | 10 May 2021 | 34 |
| AMAT | 24 May 2021 | 14 June 2021 | 18 June 2021 | 18 |
| AMAT | 23 June 2021 | 29 June 2021 | 2 July 2021 | 7 |
| AMD | 23 August 2018 | 17 September 2018 | 28 September 2018 | 25 |
| AMD | 11 November 2021 | 29 November 2021 | 1 December 2021 | 13 |

Note. Table 8 presents the date stamping result of the first five stocks. Due to the limited space, only the first five stock results are presented in Table 8.

Figure 3 shows the relationship between critical values from our Monte Carlo simulation and GSADF test statistics for each of the presented stocks. The *x*-axis represents the

year while the *y*-axis represents the GSADF test critical value. The red line is the critical value for the specific stocks for the specific date, the blue line is the GSADF test statistics, and the shaded region indicates where they intersect, suggesting potential bubble-like behavior in analyzed stocks.

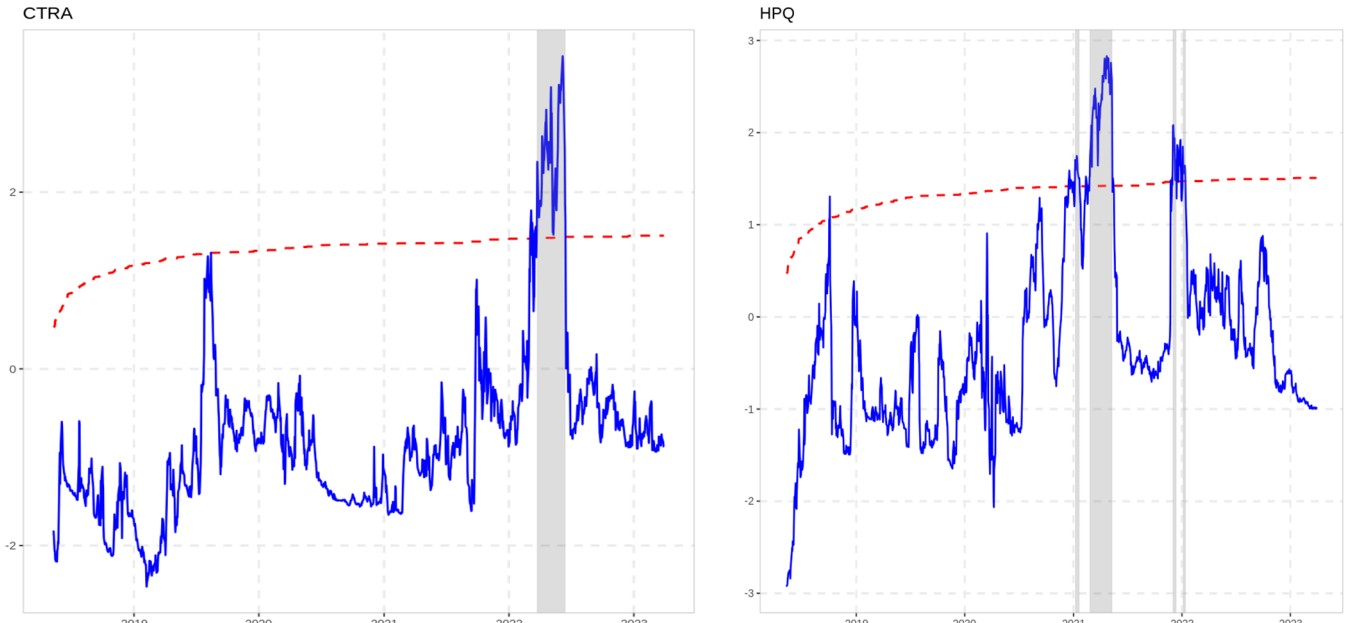

**Figure 3.** Plotting of date stamp result of CTRA and HPQ stocks. Note: Figure 3 presents the datestamp results visually for two stocks.

Figure 4 presents the number of bubble episodes during the sampled period. It reveals a notable increase in bubble episodes among 104 selected stocks from 2018 to 2023. In 2018 and 2019, there were 10 episodes each. However, in 2020, this number surged to 26, and in 2021, it further increased to 37. In contrast, 2022 saw a decline to 25 episodes, and the first quarter of 2023 had only one. In total, there were 109 bubble episodes across these stocks.

Table 9 shows the comparison of critical values obtained via SADF and GSADF approaches for each analyzed stock. For example, the first stock, AAL, yielded test statistics of −0.693 for SADF and 2.687 for GSADF. The critical values for this stock were 1.543 (SADF) and 2.387 (GSADF). As a result, the SADF test did not reject the null hypothesis, while the GSADF test rejected it at a 5% significance level.

**Table 9.** Results of SADF and GSADF tests.

| Stock | SADF Result | GSADF Result |
|---|---|---|
| AAL | Cannot reject H0 | Rejects H0 at the 5% significance level |
| AAPL | Rejects H0 at the 1% significance level | Rejects H0 at the 1% significance level |
| ABBV | Rejects H0 at the 1% significance level | Rejects H0 at the 1% significance level |
| ABT | Cannot reject H0 | Cannot reject H0 |
| AIG | Cannot reject H0 | Rejects H0 at the 1% significance level |

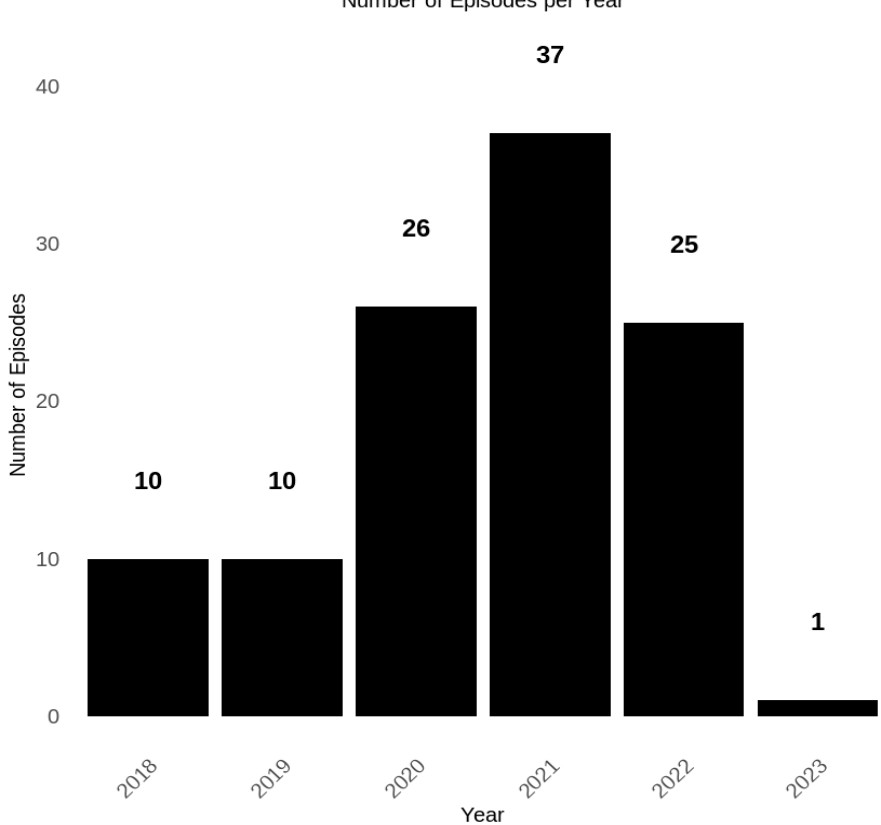

**Figure 4.** Comparison of number of bubble episodes by year.

Figure 5 displays the results of both the SADF and GSADF tests for selected stocks, indicating whether the null hypothesis (H0) is rejected at significance levels of 1%, 5%, or 10%. These tests assess whether there is a unit root, suggesting non-stationarity and a potential stock bubble. Typically, rejection at the 1% significance level includes stocks also rejected at the 5% and 10% levels, while rejection at the 5% level implies rejection at the 10% level. However, rejection at the 10% significance level does not necessarily extend to stocks rejected at the 1% and 5% levels. To maintain clarity in comparisons, this test does not aggregate stocks rejected at other significance levels.

In the "Cannot reject H0" category, 71 stocks were identified as non-rejecting using the SADF test, and 38 stocks using the GSADF test. This implies that the statistical tests failed to reject the null hypothesis at any significance level. It suggests the possibility of a unit root and non-stationarity in these stocks. These stocks do not exhibit characteristics associated with a stock bubble.

Furthermore, in the "Rejects H0 at the 1% significance level" category, 16 stocks were identified as rejecting the null hypothesis using the SADF test, while 34 stocks did so using the GSADF test. This indicates that these stocks provide strong statistical evidence against the presence of a stock bubble at the 1% significance level. Both tests strongly suggest that these stocks are unlikely to have a stock bubble and display characteristics of stationarity.

Moving to the "Rejects H0 at the 5% significance level" category, eight stocks were identified by the SADF test and fourteen stocks by the GSADF test as rejecting the null hypothesis at the 5% significance level. This implies a reduced likelihood of a stock bubble in these stocks. The results from both tests indicate that these stocks exhibit characteristics consistent with stationarity.

Finally, in the "Rejects H0 at the 10% significance level" category, nine stocks were identified using the SADF test and eighteen stocks using the GSADF test as rejecting the null hypothesis at the 10% significance level. These stocks provide evidence against a

stock bubble, though with a slightly lower level of confidence compared to the 1% and 5% significance levels.

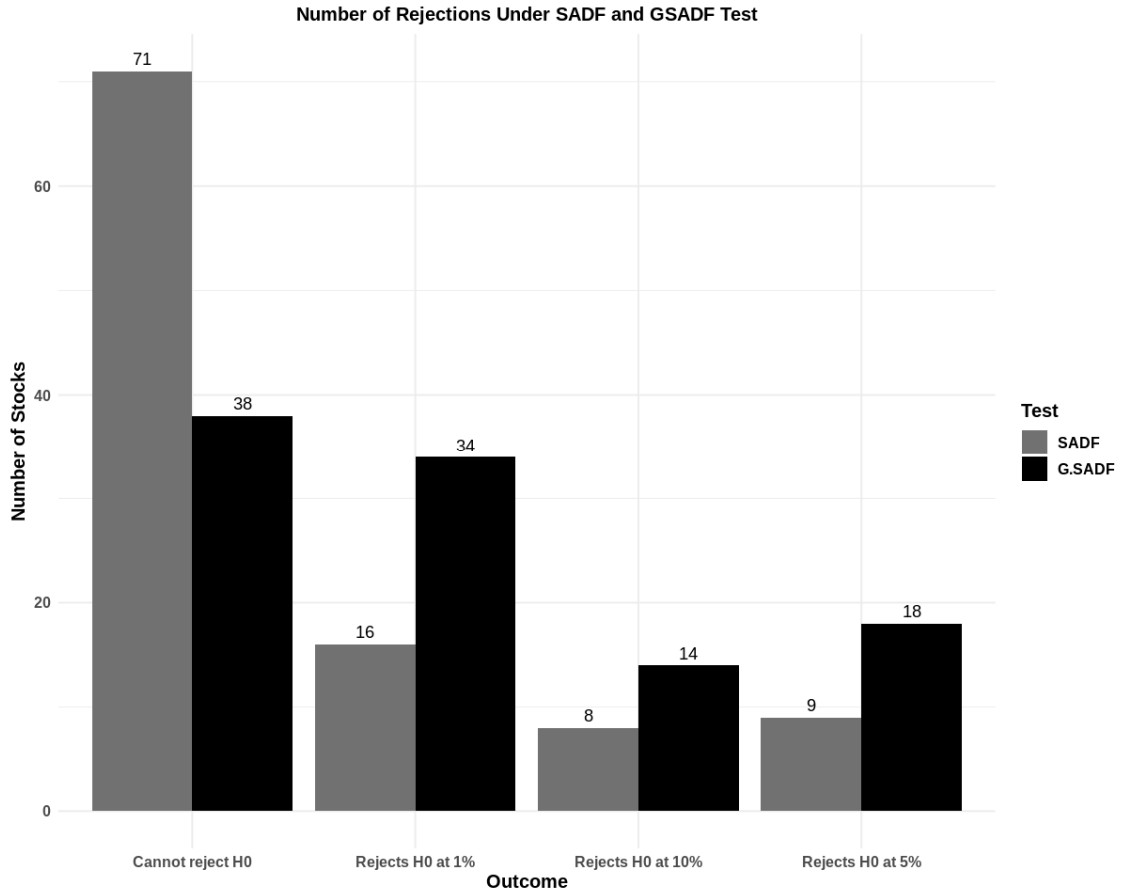

**Figure 5.** Bar graph of non-stationarity by SADF and GSADF tests.

In summary, the analysis of the outcomes reveals that the generalized supremum augmented Dickey–Fuller (GSADF) test is notably effective in detecting explosive patterns in individual stocks, as evidenced by its consistently higher rejection rates of the null hypothesis at various significance levels compared to the SADF test. Specifically, the GSADF test outperforms the SADF test by rejecting the null hypothesis for 34 stocks at the 1% level, 18 stocks at the 5% level, and 14 stocks at the 10% level, indicating a superior ability to identify potential stock bubbles. Conversely, the SADF test exhibits lower rejection rates, implying reduced sensitivity to identifying stationarity and a potentially higher risk of overlooking stock bubble occurrences. The findings strongly support the GSADF test as a more robust approach for detecting stock bubbles in individual stocks.

## 5. Conclusions

This study assessed the GSADF test's effectiveness in identifying explosive patterns in individual stocks by assessing unit roots and stationarity. The GSADF test, which examines overlapping stock price data windows for deviations from a unit root process, consistently outperformed the SADF test in rejecting the null hypothesis. It detected explosive patterns in 34 stocks at a 1% significance level, compared to 16 stocks for the SADF test. Overall, the GSADF test demonstrated greater sensitivity and effectiveness in identifying stock bubbles, while the SADF test had lower sensitivity in this regard.

Several reasons account for the GSADF test's superiority over the SADF test. Firstly, the SADF test assumes a single structural break, limiting its usefulness when multiple structural breaks or regime shifts are present. In contrast, the GSADF test is designed to handle such scenarios more effectively, making it better at detecting and quantifying

multiple structural breaks. Additionally, the SADF test can suffer from size distortions, causing the actual significance level to deviate from the desired level. This issue is addressed by the GSADF test through the use of Monte Carlo simulations, which accurately estimate significance levels, reducing size distortions and enhancing test accuracy.

While this research employs a robust econometric method endorsed by institutions like the Federal Reserve Bank of Dallas, its focus on individual stocks limits generalizability. The customization of bubble detection for specific stocks restricts systemic insights. Additionally, this study's reliance on historical daily stock data over five years excludes consideration of broader economic factors. The findings are time-dependent, and this study does not explore various contributors to bubble formation. Consequently, its applicability is constrained to the defined temporal and stock-specific parameters, cautioning against broader extrapolation or future predictions.

Since this research focuses on analyzing selected stocks' closing prices through time series data, it suggests potential avenues for future research, including exploring online learning and adaptive models for dynamic market conditions. Cross-market dynamics analysis examining interconnections among different asset classes could yield insights into bubble occurrences. Further research could explore international spillovers and contagion effects by analyzing transmission mechanisms for stock bubbles across markets and regions, considering factors like cross-border capital flows and information dissemination.

**Funding:** This research received no external funding.

**Data Availability Statement:** Publicly available datasets were analyzed in this study. This data can be found here: https://www.durgaacharya.com/p/research-project.html (accessed on 28 January 2024).

**Conflicts of Interest:** The author declares no conflicts of interest.

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
