# Peer review of "Comparative Analysis of Stock Bubble in S&P 500 Individual Stocks: A Study Using SADF and GSADF Models"

_jrfm, doi:10.3390/jrfm17020059_

Round 1
Reviewer 1 Report
Comments and Suggestions for Authors
The author(s) do not explain why bubbles in individual stocks is a major issue or topic of study. Bubbles in the general or overall stock market, overall commodities market, overall home real estate market, etc. are all subjects of intense interest. Most all of the papers cited in the paper are written on the general market bubbles. Why individual stocks? For individual stocks, it would seem that long-run abnormal returns as computed from asset pricing models is the appropriate methodology. The paper focuses on 10 highest growth and 10 lowest growth stocks. So this is like an event study of selected stocks where the event is the bubble. For this reason, I cannot accept this paper. You need to study the general stock market. If you put individual stocks in the analyses, use portfolios of the stocks and put the analyses in a subsection or separate section to complement analyses of the general stock market (S&P 500 index, NASDAQ stocks, Russell 2000 stocks, etc.).
Comments on the Quality of English LanguageNeeds some editing for English.
Author Response
Reviewer comment:
The author(s) do not explain why bubbles in individual stocks is a major issue or topic of study. Bubbles in the general or overall stock market, overall commodities market, overall home real estate market, etc. are all subjects of intense interest. Most all of the papers cited in the paper are written on the general market bubbles. Why individual stocks? For individual stocks, it would seem that long-run abnormal returns as computed from asset pricing models is the appropriate methodology. The paper focuses on 10 highest growth and 10 lowest growth stocks. So, this is like an event study of selected stocks where the event is the bubble. For this reason, I cannot accept this paper. You need to study the general stock market. If you put individual stocks in the analyses, use portfolios of the stocks and put the analyses in a subsection or separate section to complement analyses of the general stock market (S&P 500 index, NASDAQ stocks, Russell 2000 stocks, etc.).
My response:
Dear Reviewer,
Thank you for your thoughtful feedback. I am writing to provide insights into the rationale behind my decision to focus on analyzing 104 individual stocks, utilizing the GSADF method, rather than examining the entire market or a specific portfolio in my recent research.
Firstly, I would like to clarify that this research selected 104 stocks within the S&P 500, not simply the 10 highest and 10 lowest-performing stocks as mentioned in your comments. I apologize if this was unclear in the paper. By analyzing individual stocks, I believe I can provide readers, particularly retail traders, with a more granular understanding of how bubbles emerge, how to detect them, and how to potentially benefit from bubble-like behavior. I recognize that not all stock traders are interested in the broader market picture, but rather focus on specific stocks they own or trade, and would appreciate insights regarding their individual performance.
The chosen 104 stocks met specific criteria:
- S&P 500 Inclusion: All stocks belong to the S&P 500 index.
- Volatility Threshold: They experienced at least a 10% price increase in a single day over the five-year period from January 1, 2018, to March 31, 2023. This threshold aimed to select stocks prone to higher volatility compared to the overall market, potentially indicating bubble tendencies.
- Trading Volume: Each stock had a minimum average daily trading volume of one million. This criterion aimed to mitigate the impact of low liquidity on price volatility and ensure the chosen stocks reflected market sentiment more accurately.
The reasoning behind these criteria is as follows:
- Market Representation: Focusing on the entire market might not effectively capture bubble dynamics due to averaging effects and potential biases based on stock weights. Selecting individual stocks with high volatility helps isolate potential bubble behavior.
- Volatility Threshold: By choosing stocks with significant single-day price jumps, we focus on those exhibiting potentially excessive exuberance, increasing the likelihood of bubble presence.
- Trading Volume: Low-volume stocks are susceptible to greater price fluctuations due to limited liquidity. Choosing stocks with higher trading volume ensures the observed price movements reflect broader market sentiment and are less prone to manipulation.
In regards to the model selection, previous studies on the detection of asset price bubbles have mainly relied on identifying specific historical events and attempting to predict the likelihood of future occurrences. However, recent advancements in right-tailed unit root tests have provided new tools for real-time monitoring and surveillance of asset prices and the SADF and GSADF test have demonstrated remarkable accuracy in identifying both the emergence and subsequent collapse of bubbles.
The GSADF test, in particular, has gained traction as a reliable early warning system for bubble-like behavior across various financial markets. Its effectiveness is exemplified by its adoption by prominent institutions like the Federal Reserve Bank of Dallas, which uses it to monitor potential exuberance in international housing markets.
Therefore, by employing a GSADF test, I aimed to leverage a data-driven, real-time approach to bubble detection. This innovative methodology represents a significant step forward in understanding stock bubble behavior and the associated risks of asset price bubbles.
I hope this clarifies my research design and addresses your concerns. Please do not hesitate to reach out if you have any further questions.
Thank you for your time and valuable feedback.
Sincerely,
Dr. Durga
Reviewer 2 Report
Comments and Suggestions for Authors
Please see attached report

Excellent use of English language but some problems remain, see attached report
Author Response
Reviewer comments:
Excellent use of English language but some problems remain, see attached report.
My Response
Dear Reviewer,
I am writing to express my sincere gratitude for your insightful and positive feedback on my paper. Your comments have been invaluable in guiding me to refine and strengthen my work. I have carefully reviewed all the points you raised in your report and have implemented revisions to address each concern including defining the variables, grammatical corrections, and so forth.
In particular, I recognize your observation about the lack of limitations explicitly mentioned in the conclusion. I have rectified this by adding a concise statement acknowledging the study's limitations. Additionally, I want to clarify that recommendations for future research are indeed included in the final paragraph of the conclusion section.
I am currently working on incorporating the revisions and expect to submit the updated manuscript once it becomes available on the portal.
Regards
Dr. Durga
Round 2
Reviewer 1 Report
Comments and Suggestions for Authors
None.
Comments on the Quality of English LanguagePlease work to edit manuscript and writing English language to some extent.